# Identification of regulatory targets for the bacterial Nus factor complex

Gabriele Baniulyte[1,2], Navjot Singh[1], Courtney Benoit[1], Richard Johnson[1,2], Robert Ferguson[1], Mauricio Paramo [1], Anne M. Stringer[1], Ashley Scott[1], Pascal Lapierre[1] & Joseph T. Wade[1,2]

Nus factors are broadly conserved across bacterial species, and are often essential for viability. A complex of five Nus factors (NusB, NusE, NusA, NusG and SuhB) is considered to be a dedicated regulator of ribosomal RNA folding, and has been shown to prevent Rho-dependent transcription termination. Here, we identify an additional cellular function for the Nus factor complex in *Escherichia coli*: repression of the Nus factor-encoding gene, *suhB*. This repression occurs primarily by translation inhibition, followed by Rho-dependent transcription termination. Thus, the Nus factor complex can prevent or promote Rho activity depending on the gene context. Conservation of putative NusB/E binding sites upstream of Nus factor genes suggests that Nus factor autoregulation occurs in many bacterial species. Additionally, many putative NusB/E binding sites are also found upstream of other genes in diverse species, and we demonstrate Nus factor regulation of one such gene in *Citrobacter koseri*. We conclude that Nus factors have an evolutionarily widespread regulatory function beyond ribosomal RNA, and that they are often autoregulatory.

[1] Wadsworth Center New York State Department of Health, Albany, NY 12208, USA. [2] Department of Biomedical Sciences, School of Public Health University at Albany, Rensselaer, NY 12144, USA. Correspondence and requests for materials should be addressed to J.T.W. (email: joseph.wade@health.ny.gov)

Nus factors are widely conserved in bacteria and play a variety of important roles in transcription and translation[1]. The Nus factor complex comprises the four classical Nus factors, NusA, NusB, NusE (ribosomal protein S10), NusG, and a recently discovered member, SuhB. As a complex, Nus factors serve an important role in promoting expression of ribosomal RNA (rRNA)[2,3]. A NusB/E complex binds BoxA sequence elements in nascent rRNA, upstream of the 16S and 23S genes[4,5]. Once bound to BoxA, NusB/E has been proposed to interact with elongating RNAP via the NusE–NusG interaction[6]. The role of NusA in Nus complex function is unclear, but may involve binding of NusA to RNA flanking the BoxA[7]. NusA has also been proposed to be a general Rho antagonist by competing with Rho for RNA sites[8]. Early studies of Nus factors focused on their role in preventing both Rho-dependent and intrinsic termination of λ bacteriophage RNAs ('antitermination')[9], which is completely dependent on the bacteriophage protein N. Nus factors can prevent Rho-dependent termination in the absence of N[10,11], and for many years, Nus factors were believed to prevent Rho-dependent termination of rRNA[9]. However, it was recently shown that rRNA is intrinsically resistant to Rho termination, and that the primary role of Nus factors at rRNA is to promote proper RNA folding during ribosome assembly[3,12].

The most recently discovered Nus factor, SuhB, has been proposed to stabilize interactions between the NusB/E-bound BoxA and elongating RNAP, thus contributing to proper folding of rRNA[12]. Genome-wide approaches revealed that suhB is upregulated in the presence of the Rho inhibitor bicyclomycin, suggesting that suhB is subject to premature Rho-dependent transcription termination[13,14]. Surprisingly, suhB is also one of the most upregulated genes in ΔnusB cells[12], suggesting a possible autoregulatory function for Nus factors. Moreover, autoregulation of suhB has been suggested previously[15], although the mechanism for this regulation is unclear.

Here, we show that suhB is translationally repressed by Nus factors, which in turn leads to premature Rho-dependent transcription termination. This represents a novel mechanism for control of premature Rho-dependent termination, and is the first described cellular function for Nus factors beyond regulation of rRNA. Moreover, the role of Nus factors at suhB is to promote Rho-dependent termination of suhB, in contrast to their established function in antagonizing Rho. Bioinformatic analysis suggests that regulation by Nus factors is widespread, and that autoregulation of genes encoding Nus factors, in particular suhB, is a common phenomenon. We confirm Nus factor association with suhB mRNA in Salmonella enterica, and we demonstrate Nus factor regulation of an unrelated gene in Citrobacter koseri. Thus, our data show that Nus factors are important regulators with diverse targets and diverse regulatory mechanisms.

## Results

**Rho-dependent termination within the suhB gene**. Genome-wide analysis of Rho termination events suggested Rho-dependent termination within the Escherichia coli suhB gene[13,14]. To confirm this, we used Chromatin Immunoprecipitation (ChIP) coupled with quantitative PCR (ChIP-qPCR) to determine RNAP association across the suhB gene in wild-type cells and cells expressing a mutant Rho (R66S) that has impaired termination activity, likely due to a defect in RNA loading[16]. In wild-type cells, we observed a large decrease in RNAP association at the 3′ end of suhB relative to the 5′ end. This decrease was substantially reduced in rho mutant cells (Fig. 1). Thus, our ChIP data independently support the observation of Rho termination within suhB[13,14].

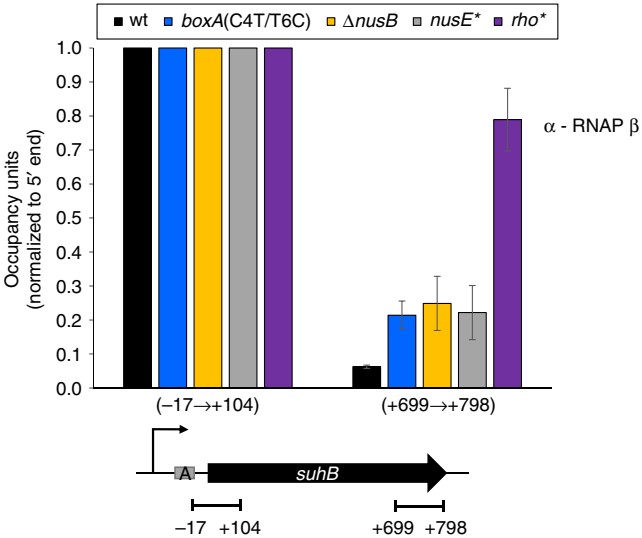

**Fig. 1** Transcription termination within suhB is dependent on Rho and Nus factors. RNAP (β) enrichment at suhB 5′ and 3′ regions was measured using ChIP-qPCR in wild-type MG1655, boxA(C4T/T6C), ΔnusB, nusE(A12E) or rho(R66S) mutant strains. Values are normalized to signal at the 5′ end of suhB. x-axis labels indicate qPCR amplicon position relative to suhB. Error bars represent ±1 standard deviation from the mean (n = 3). A schematic depicting the suhB gene, the transcription start site (bent arrow) and boxA (grey rectangle) is shown below the graph. Horizontal black lines indicate the position of PCR amplicons

**Nus factors are trans-acting regulators of suhB**. Based on an approach used to identify modulators of Rho-dependent termination within S. enterica chiP[17], we used a genetic selection to isolate 30 independent mutants defective in Rho-dependent termination within suhB (see Methods). All 30 strains isolated had a mutation in one of three genes: nusB (14 mutants), nusE (13 mutants) or nusG (3 mutants) (Supplementary Table 1). We then measured RNAP association across the suhB gene in wild-type, ΔnusB and nusE mutant cells (nusE A12E mutant isolated from the genetic selection). Mutation of nusB or nusE increased RNAP binding at the suhB 3′ end ~4-fold compared to wild-type cells (Fig. 1 and Supplementary Fig. 1). We conclude that Nus factors promote Rho-dependent termination within the suhB gene. However, RNAP occupancy at the 3′ end of suhB in nusB and nusE mutants was substantially lower than in the rho mutant (Fig. 1 and Supplementary Fig. 1). This difference may be due to spurious, non-coding transcripts arising from nearby intragenic promoters, which are widespread in E. coli[18] and are often terminated by Rho[13,14].

**A functional BoxA in the suhB 5′ UTR**. We identified a sequence in the suhB 5′ UTR with striking similarity to boxA sequences from rRNA loci (Supplementary Table 2). Moreover, this boxA-like sequence is broadly conserved across Enterobacteriaceae species (Fig. 2a and Supplementary Fig. 2), suggesting that it is a genuine binding site for NusB/E. We generated a library of randomly mutated suhB-lacZ transcriptional fusions (see Methods), and identified fusions that had higher expression of lacZ. All identified mutants carried a single nucleotide change at one of five different positions within the putative boxA (Fig. 2b). We then constructed a strain carrying two chromosomal point mutations in the putative suhB boxA (C4T/T6C; numbers corresponding to the position in the consensus boxA; Supplementary Table 2). We used ChIP-qPCR to measure association of FLAG-tagged SuhB at the 5′ end of the suhB gene in

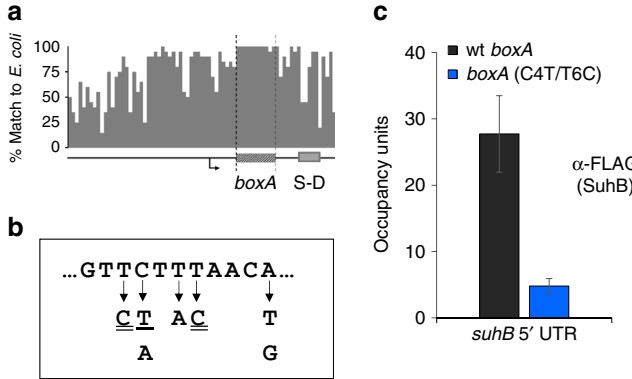

**Fig. 2** A functional BoxA in the 5′ UTR of *suhB*. **a** Sequence conservation of the 100 bp upstream of *suhB* and its homologues across 20 *Enterobacteriaceae* species. The transcription start site is indicated by a bent arrow, and the BoxA and S-D sequences are indicated. **b** List of *boxA* mutations that are associated with increased *suhB* expression. All single nucleotide changes are indicated by an arrow. Single underline indicates a mutation that was isolated in the absence of mutations anywhere else in the cloned region; other mutants included additional mutations outside the *boxA*. Double underline indicates that the *boxA* mutation was isolated in two or more independent clones. Critical position '−4' is indicated (see Supplementary Table 2). **c** SuhB association with the 5′ end of *suhB* in wild-type ('wt') and *boxA* mutant ('boxA C4T/T6C') strains. SuhB-FLAG occupancy was measured by ChIP-qPCR using α-FLAG antibody. Error bars represent ±1 standard deviation from the mean ($n = 3$)

wild-type cells, or cells containing the *boxA* mutation. We detected robust association of SuhB-FLAG in wild-type cells, but not in the *boxA* mutant strain (Fig. 2c). We conclude that the putative BoxA in the 5′ UTR of *suhB* is genuine, and recruits Nus factors. To test whether the BoxA controls Rho-dependent termination within *suhB*, we measured RNAP occupancy across *suhB* in the *boxA* mutant strain. We detected a ~4-fold increase in RNAP occupancy in the downstream portion of *suhB* in the *boxA* mutant strain relative to wild-type cells, mirroring the effect of mutating *nusB* or *nusE* (Fig. 1). Our data support a model in which Nus factor recruitment by the *suhB* BoxA leads to Rho-dependent termination within the gene.

**Nus factors mediate translational repression of *suhB*.** The *suhB* BoxA is separated by only 6 nt from the Shine-Dalgarno (S-D) sequence (Fig. 2a). Rho cannot terminate transcription of translated RNA, likely because RNAP-bound NusG interacts with ribosome-associated NusE (S10)[6]. Hence, we hypothesized that NusB/E association with BoxA sterically blocks association of the 30S ribosome with the mRNA, repressing translation initiation, uncoupling transcription and translation, and thereby promoting Rho-dependent termination. To test this hypothesis, we used the *suhB-lacZ* transcriptional fusion (Fig. 3a), as well as an equivalent translational fusion (Fig. 3b). We reasoned that mutation of *nusB*, *nusE* or *boxA* would result in increased expression from both reporter fusions, since these mutations would relieve translational repression (reported by the translational fusion), which in turn would reduce Rho-dependent termination (reported by the transcriptional fusion). In contrast, we reasoned that mutation of *rho* would result in increased expression only from the transcriptional fusion reporter, since the SuhB-LacZ fusion protein (from the translational fusion construct) would still be translationally repressed. We measured expression of *lacZ* from each of these reporter fusions in wild-type cells, and cells with Δ*nusB*, *nusE* A12E or *rho* R66S mutations. We also measured expression of *lacZ* in these strains using reporter fusions carrying the C4T/

T6C *boxA* mutation. Consistent with our model, we detected increased expression of both reporter fusion types in mutants of *nusB*, *nusE* or *boxA*, whereas mutation of *rho* resulted in increased expression of the transcriptional fusion but not the translational fusion reporter (Fig. 3a, b). Note that mutation of *nusB*, *nusE* or *boxA* does not lead to the same level of increase in expression of the reporter fusions (Fig. 3a, b). This is likely due to the fact that mutations in Nus factors have extensive pleiotropic effects, presumably due to the importance of Nus factors in ribosome assembly[3]. Moreover, mutation of *boxA* in a *nusE* mutant leads to a further increase in reporter expression, whereas mutation of *boxA* in a *nusB* mutant does not (Fig. 3a, b). This is likely due to the mutant NusE retaining partial function, whereas deletion of *nusB* completely abolishes Nus factor function.

To confirm the effects of mutating *nusB*, *nusE*, *rho* and *boxA* on expression of *suhB* in the native context, we measured SuhB protein levels by western blotting using strains expressing a C-terminally FLAG-tagged derivative of SuhB. We compared SuhB protein levels in cells with *nusE* A12E, *rho* R66S or *boxA* C4T/T6C mutations; we have previously shown that SuhB protein levels are increased in a Δ*nusB* mutant[12]. SuhB protein levels in the mutant strains correlated well with the translational *suhB-lacZ* fusion reporter gene assay: mutation of *nusE* or *boxA* caused a modest increase in SuhB-FLAG levels, whereas mutation of *rho* had no discernible effect (Fig. 3c, d; Supplementary Fig. 7).

**Rho-dependent transcription termination occurs early in the *suhB* gene.** Rho-dependent termination requires a Rho loading sequence known as a Rut that typically occurs >60 nt upstream of the termination site(s), is pyrimidine-rich and G-poor[19]. To localize the Rut and the downstream termination site(s), we constructed a short transcriptional *suhB-lacZ* fusion that includes only the first 57 bp of the *suhB* gene. Expression of this reporter fusion was substantially higher in *rho* mutant cells than in wild-type cells (Fig. 4a). In contrast, expression was only marginally higher in *rho* mutant cells than in wild-type cells when the *boxA* sequence was mutated (Fig. 4a). Thus, the short *suhB-lacZ* reporter fusion behaves similarly to the fusion that includes the entire *suhB* gene (Fig. 3a), indicating that the *rut* and termination sequences must be upstream of position 57 within *suhB*.

Given that the short *suhB-lacZ* fusion includes only 94 bp of transcribed sequence from *suhB* and its 5′ UTR, and that Rut sequences are typically found >60 nt from the site(s) of termination[19], we reasoned that the Rut is likely located close to the 5′ end of the *suhB* 5′ UTR. Consistent with this, positions 2–22 of the 5′ UTR include 17 pyrimidines and only one G. This sequence completely encompasses the *boxA*, suggesting that the *boxA* and *rut* sequences overlap. To determine whether mutation of the *boxA* affects Rho-dependent termination independent of Nus factor-mediated translational repression, we constructed short *suhB-lacZ* fusions in which the *suhB* start codon was mutated, either alone or in conjunction with a mutated *boxA*. We reasoned that mutation of the *suhB* start codon would bypass the need for BoxA-mediated translational repression to cause Rho-dependent termination. As expected, expression of the fusion with the mutated start codon but wild-type *boxA* was substantially higher in a *rho* mutant than in wild-type cells (Fig. 4a), consistent with this construct being Rho-terminated. However, expression of the fusion with the mutated start codon and mutated *boxA* was only marginally higher in a *rho* mutant than in wild-type cells, indicating that Rho-dependent termination is disrupted by mutation of the *boxA*, even in the absence of *suhB* translation. We conclude that mutation of the *boxA* reduces Rho-dependent termination by disrupting the *rut*. This likely occurs due to the *boxA* and *rut* sequences overlapping, in which case

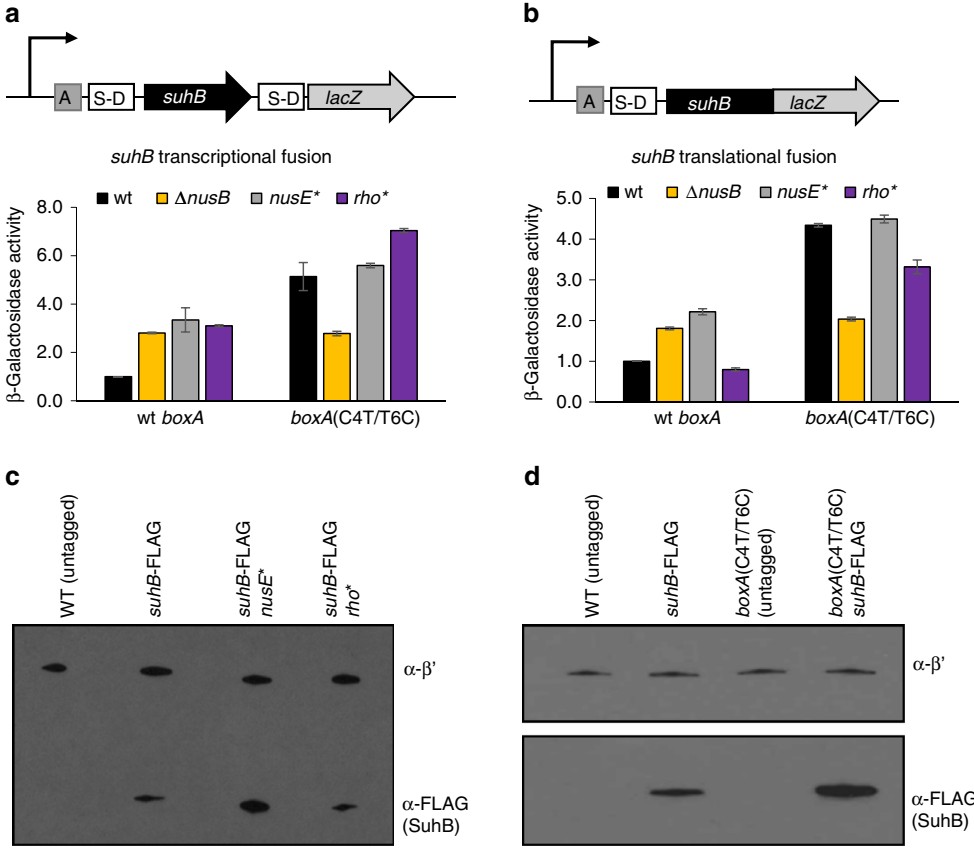

**Fig. 3** Nus factors repress translation of *suhB*, leading to Rho-dependent termination within the gene. β-galactosidase activity of (**a**) transcriptional and (**b**) translational fusions of *suhB* to *lacZ* in wild-type cells, Δ*nusB*, *nusE*(A12E), or *rho*(R66S) mutants. The *suhB-lacZ* fusion had either a wild-type ('wt') or mutant *boxA* ('C4T/T6C'). Data are normalized to levels in wild-type cells. Error bars represent ±1 standard deviation from the mean (*n* = 3). Schematics of constructs used in these experiments are depicted above the graphs. (**c**) and (**d**) Western blots showing SuhB-FLAG protein levels in wild-type cells, *nusE* (A12E), *rho*(R66S) **c**, and *boxA*(C4T/T6C) mutants **d**. SuhB-FLAG was probed with α-FLAG antibody; RNAP β' was probed as a loading control. Representative blots from at least three independent experiments are shown. Images were cropped for clarity; unprocessed western blot images are in Supplementary Fig. 7

mutating the *boxA* would also alter the *rut*. However, mutation of the *boxA* might also alter RNA secondary structure of the Rut.

Mutation of the *boxA* results in greatly decreased Rho-dependent termination of a fusion of the entire *suhB* gene to *lacZ* (Fig. 3a). Although this effect could be due to disruption of a *rut* overlapping the *boxA*, we reasoned that there are likely to be additional *rut* sequences within the *suhB* ORF. To test this hypothesis, we constructed transcriptional fusions of the entire *suhB* gene and 5′ UTR to *lacZ* with a mutation in the *suhB* start codon, either alone or in conjunction with a mutation in the *boxA*. For both constructs, expression was substantially higher in a *rho* mutant than in wild-type cells (Fig. 4b), indicating robust Rho-dependent termination within the *suhB* gene, even with a mutated *boxA*. We conclude that the *suhB* gene includes at least one additional *rut*, and that the effect of mutating the *boxA* on Rho-dependent termination with a long transcriptional fusion (Fig. 3) is due to loss of Nus factor binding rather than a direct effect on Rho loading.

**BoxA-mediated occlusion of the S-D sequence is not due to steric occlusion.** The data described above are consistent with a steric occlusion model in which NusB/E binding to the BoxA directly prevent 30S ribosome association with the S-D sequence. However, other mechanisms of translational repression are also possible. The steric occlusion model predicts that increasing the

distance between the *boxA* and S-D elements would relieve translational repression, and consequently Rho-dependent termination. We constructed *suhB-lacZ* transcriptional fusions that carried insertions of sizes from 2 to 100 bp between the *boxA* and S-D sequences (see Methods for details). We constructed equivalent fusions carrying a *boxA* mutation (C4A; Supplementary Table 2). Surprisingly, separating the BoxA and S-D sequences with up to 100 nt intervening RNA did not abolish BoxA-mediated repression (Fig. 5). Note that differences in absolute expression levels for the different constructs are likely due to variability in secondary structure around the ribosome binding site. Additionally, we are confident that none of the insertions inadvertently introduces a new promoter, since a similar construct lacking an active upstream promoter was only weakly expressed (Supplementary Fig. 3). We conclude that the steric occlusion model is insufficient to explain BoxA-mediated translational repression of *suhB*, although the proximity of the BoxA and S-D sequences suggests that simple occlusion would prevent ribosome binding.

We reasoned that if steric occlusion of ribosomes by NusB/E binding is sufficient for repression of *suhB*, it would not require assembly of a complete Nus factor complex, since NusB/E alone has a high affinity for BoxA RNA[4]. Hence, we constructed *suhB-lacZ* translational fusions where the native promoter is replaced by a T7 promoter. Previous studies showed that gene regulation

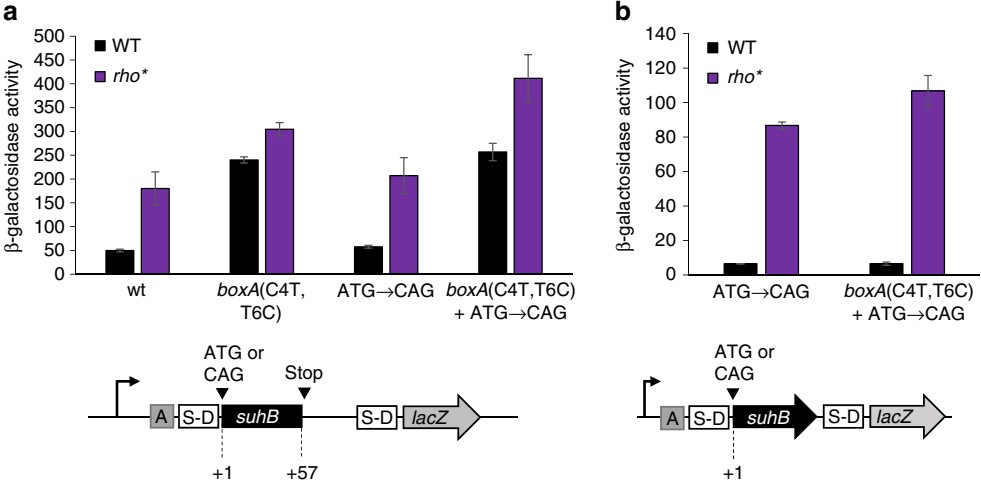

**Fig. 4** The *suhB* BoxA overlaps the first of multiple Rut elements. β–galactosidase activity of short (**a**) and full-length (**b**) *suhB* transcriptional fusions to *lacZ*. Constructs included either a wild-type sequence ('wt'), *boxA* mutation ('C4T, T6C') and/or *suhB* start codon mutation ('ATG → CAG'), as indicated on the *x*-axis. 200 bp of sequence upstream of the *suhB* gene was included in all constructs. Error bars represent ±1 standard deviation from the mean (*n* = 3). Schematics of the constructs used for these experiments are depicted below the graphs

involving λ N or NusG is lost when *E. coli* RNAP is substituted with bacteriophage T7 RNAP[20–22], suggesting that T7 RNAP does not interact with Nus factors; hence, transcription of this *suhB-lacZ* fusion by T7 RNAP would not be associated with the formation of a complete Nus factor complex. We grew cells at 37 °C, 30 °C or room temperature (23 °C), since the transcription elongation rate of T7 RNAP is similar to that of *E. coli* RNAP at room temperature, but considerably higher at 37 °C[23,24]. At all temperatures, we detected robust expression that was dependent upon expression of T7 RNAP in the same cells. However, we observed no effect on expression of mutating the *boxA* (Supplementary Fig. 4). We conclude that efficient BoxA-dependent repression of *suhB* requires assembly of a complete Nus factor complex.

**Salmonella enterica suhB has a functional BoxA**. Phylogenetic analysis of the region upstream of the *suhB* gene indicates that the *boxA* sequence is widely conserved among members of the family *Enterobacteriaciae* (Fig. 2a; Supplementary Table 2; Supplementary Fig. 2), suggesting that BoxA-mediated regulation of *suhB* occurs in these species. To investigate this possibility, we used ChIP of FLAG-tagged SuhB to measure association of SuhB with the *suhB* upstream region in *S. enterica* subspecies *enterica* serovar Typhimurium. We detected robust association of both RNAP (β subunit) and SuhB with the *suhB* upstream region (Supplementary Fig. 5), indicating that the *suhB* mRNA contains a functional BoxA. We also failed to detect association with a previously reported cryptic BoxA within the *hisG* gene (Supplementary Fig. 5), consistent with the sequence of this element differing at a critical position from the BoxA consensus (Supplementary Table 2).

**BoxA-mediated regulation and Nus factor autoregulation are phylogenetically widespread**. Aside from their role in lambdoid phage, Nus factors have historically been considered dedicated regulators of rRNA expression. Our discovery of *suhB* as a novel regulatory target of Nus factors suggests that BoxA-mediated regulation may be more extensive. BoxA sequences in rRNA are known to be highly conserved[2]. Based on the *boxA* sequences from *E. coli* rRNA and *suhB* loci, and a previous analysis of sequences required for BoxA function in *E. coli*[5], we derived a consensus sequence (GYTCTTTAANA) that is likely to be

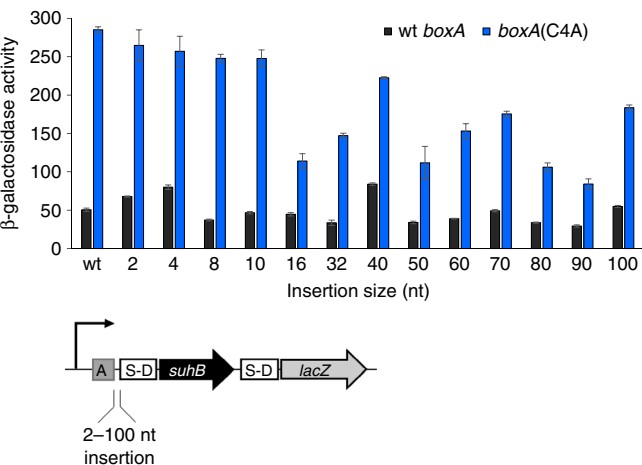

**Fig. 5** The effect on *suhB-lacZ* transcription levels of altering the distance between *boxA* and the S-D sequence. β–galactosidase activity of wild-type ('wt *boxA*'; dark grey bars) and *boxA* mutant ('C4A'; blue bars) transcriptional fusions of *suhB* to *lacZ*, with increasing lengths of non-coding DNA inserted between the *boxA* and S-D sequences. The length of inserted sequence (nt) is indicated on the *x*-axis. Constructs include 200 bp of upstream sequence and a full-length *suhB* fused to *lacZ* in the pAMD-BA-lacZ plasmid. Note that the sequence of inserted non-coding DNA differs for constructs with insertion sizes of ≤32 bp and ≥40 bp (see Methods and Supplementary Fig. 6 for details)

applicable to almost all γ-proteobacteria[2]. We searched for perfect matches to this sequence in 940 sequenced γ-proteobacterial genomes. We then selected sequence matches that are positioned within 50 bp of a downstream start codon for an annotated gene. Thus, we identified 407 putative BoxA sequences from 314 genomes, with between 0 and 7 instances per genome (Supplementary Data 1). We determined whether any gene functions were identified from multiple genomes. To minimize biases from the uneven distribution of genome sequences across different genera, we analysed gene functions at the genus rather than species level. Across all the species analysed, we identified 36 different gene functions with at least one representative from one genus. Strikingly, we identified 34 of 55 genera in which at least one species has a putative *boxA* sequence within 50 bp of the start of

an annotated *suhB* homologue. We identified three additional genera in which at least one species has a putative *boxA* within 50 bp of the start of an unannotated *suhB* homologue, and one genus with a species in which the *suhB* homologue has a putative *boxA* 82 bp from the gene start. Thus, our analysis reinforces the notion that BoxA-mediated regulation of *suhB* is highly conserved (Fig. 2a and Supplementary Fig. 2). Three other gene functions were represented in multiple genera: *prsA* (encodes ribose-phosphate pyrophosphokinase) and *rpsJ* (encodes NusE) were each found in three genera, and genes encoding ParE-like toxins were found in two genera. We also identified two genera with species in which *rpsJ* is predicted to be a downstream gene in an operon where the first gene in the operon has a putative *boxA* < 50 bp from the gene start.

**BoxA-mediated regulation of a toxin-antitoxin system in Citrobacter koseri.** Bioinformatic analysis strongly suggested that BoxA-mediated regulation is evolutionarily widespread and extends to genes other than *suhB*. To determine whether Nus factors regulate genes other than rRNA and *suhB* in other species, we selected one putative BoxA-regulated gene identified by the bioinformatic search for *boxA*-like sequences: *CKO_00699* from *C. koseri* (Supplementary Table 2). *CKO_00699* is predicted to encode a ParE-like toxin, part of a putative toxin−antitoxin pair. A putative *boxA* was observed upstream of a homologous gene in *Pasteurella multocida*, suggesting conserved BoxA-mediated regulation. We reasoned that if *CKO_00699* is a genuine target of Nus factors, it would likely retain this regulation in *E. coli*, since Nus factors are highly conserved between *C. koseri* and *E. coli* (e.g. the amino acid sequence of NusB is 97% identical and 100% similar between the two species). Hence, we constructed a transcriptional fusion of *CKO_00699* to *lacZ* and measured expression in *E. coli*. Note that we included a mutation in *CKO_00699* (R82A) to inactivate the predicted toxin activity to prevent growth inhibition. The *lacZ* fusion included a strong, constitutive promoter[25], and the sequence from *C. koseri* began at the predicted transcription start site, based on manual analysis of likely promoter sequences (Fig. 6). We measured expression of fusions with wild-type and mutant *boxA* (C4A) sequences (Supplementary Table 2), in wild-type and Δ*nusB* strains. Mutation of the putative *boxA*, or deletion of *nusB* resulted in a substantial increase in expression, whereas mutation of the *boxA* did not affect expression in the Δ*nusB* strain (Fig. 6). We conclude that *CKO_00699* is directly repressed by a BoxA and Nus factors.

## Discussion

We have shown that premature Rho-dependent termination within the *suhB* gene is controlled by a BoxA and Nus factors. This likely serves as a mechanism for autoregulation of Nus factors, since SuhB is a critical component of the Nus machinery[12]. Premature Rho-dependent termination of mRNAs has been recently recognized to be a widespread regulatory mechanism[26,27]. Most regulation of this type occurs by alteration of mRNA accessibility around Rut sites. In the case of *suhB*, Rho-dependent termination occurs as a result of translational repression.

A function for Nus factors in promoting Rho-dependent termination is particularly striking because of their long association with antitermination[9]. The contrasting effects of Nus factors on Rho-dependent termination in different contexts, and their role in promoting ribosomal assembly, highlight the flexibility in the function of these proteins. Our data indicate that translational repression of *suhB* by Nus factors is not due to occlusion of the S-D. Previous studies of Nus factors suggest that they form a loop between the BoxA in the RNA and the elongating RNAP[3,12]. We

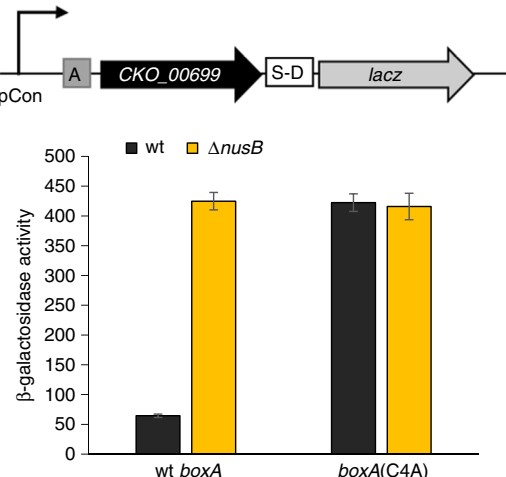

**Fig. 6** Identification of a BoxA element in *Citrobacter koseri*. β-galactosidase activity of wild-type ('wt *boxA*') and *boxA* mutant ('C4A') transcriptional fusions of *CKO_00699* (R82A mutant, to avoid potential toxicity to *E. coli* in the absence of the anti-toxin) to *lacZ* in *E. coli* wild-type ('wt'; dark grey bars) or *nusB* deletion ('Δ*nusB*'; yellow bars) strains. *CKO_00699-lacZ* transcription was driven by a constitutive promoter[25]. Error bars represent ±1 standard deviation from the mean ($n = 3$)

propose that this loop prevents the 30S ribosome from accessing the S-D. Alternatively, association of NusG with NusE in the context of the Nus factor complex may prevent translation by blocking association of NusG with ribosome-associated NusE (S10).

Autoregulation of SuhB is strikingly similar to autoregulation of λ N. λ *nutL* is positioned ~200 bp upstream of the *N* gene. Binding of Nus factors and N to NutL results in translational repression of N[28]. The distance between NutL and the S-D sequence is such that a simple steric occlusion model is insufficient to explain translational repression by N and Nus factors; the RNA loop formed between NutL and the elongating RNAP provides a straightforward explanation of repression. Although the gap between NutL and the S-D sequence for the *N* gene is considerably longer than the longest distance we tested for *suhB* (Fig. 5), the intervening sequence is highly structured[29], which may impact the compactness of the loop.

Although we have shown previously that Nus factors are not required to prevent Rho-dependent termination at rRNA loci[12], Nus factors have been shown to prevent Rho-dependent termination in artificial reporter constructs[10,11,30,31]. Our finding that Nus factors promote Rho-dependent termination in *suhB* further indicates that context determines the precise function of Nus factors. Hence, it is likely that there are additional sequence elements in *suhB* that promote Rho-dependent termination, or that there are additional sequence elements in the artificial reporter constructs that prevent Rho-dependent termination.

Our data support a widespread regulatory role for Nus factors, implicating them in regulation in both a wide range of species, and of a diverse set of genes, although within any given species there are likely only a few regulatory targets. Strikingly, ~25% of the gene functions associated with an upstream *boxA* are known to be directly connected to translation. This is consistent with the established connection between Nus factors and ribosomal assembly[3], and suggests that the impact of Nus factors on translation occurs by regulation of a variety of genes. Moreover, our data suggest that NusE is autoregulated in phylogenetically diverse species. Although we did not identify any genomes where genes encoding other Nus factors have putative upstream *boxA* sequences, we did identify a putative *boxA* sequence upstream of

*ribH* in six different species of *Pseudomonas*. In all cases, *nusB* is the gene immediately downstream of *ribH*, suggesting that *nusB* is autoregulated in pseudomonads. Overall, we identified no species with a putative *boxA* upstream of more than one Nus factor-encoding gene, and only 11 genera had no putative *boxA* associated with any Nus factor-encoding gene. However, for five of these latter genera we were unable to identify a *boxA* sequence upstream of the rRNA genes, suggesting that the BoxA consensus is different to that in *E. coli*. Thus, our data strongly suggest that Nus factor autoregulation occurs in ~90% of gamma-proteobacterial species, and that typically, just one Nus factor is autoregulated. The evidence for autoregulation of SuhB, NusE and NusB suggests that the levels of these proteins contribute to feedback loops that control the primary function of Nus factors: promoting ribosomal assembly. Our observation of BoxA-mediated regulation of a ParE-like toxin in *C. koseri* demonstrates that Nus factors regulate genes other than their own. Indeed, our bioinformatic analysis suggests that genes of many functions may be regulated by Nus factors, with 36 gene functions represented in at least one genus. Our list is conservative because (i) it does not consider the possibility of regulation by BoxA sequences located >50 nt upstream of the gene start, which we know is possible (Fig. 5), (ii) it does not consider non-coding RNAs, (iii) the BoxA consensus may be different in some of the species analysed, and (iv) gene starts predicted by bioinformatic annotation pipelines may be incorrect[32].

Our data indicate that regulation by Nus factors extends to many genes beyond rRNA, and that Nus factor autoregulation is an evolutionarily widespread phenomenon. Moreover, we have shown that Nus factors can provide contrasting forms of regulation, depending on the context of the target; despite their long-established function in antitermination[9], Nus factors promote Rho-dependent termination within *suhB*. Key questions about the function of Nus factors remain to be addressed. What is the molecular architecture of the Nus factor machinery? What are the specific RNA sequences that determine whether Nus factors prevent Rho-dependent termination? How do Nus factors modulate the function of elongating RNAP? Our identification of novel Nus factor target genes with novel regulatory mechanisms provides an excellent opportunity to address these questions.

## Methods

**Strains and plasmids**. All strains, plasmids and oligonucleotides used in this study are listed in Supplementary Tables 3 and 4. Mutations in *rpsJ* and *rho* were P1 transduced into MG1655[33] MG1655 Δ*lacZ* (AMD054)[34] and MG1655*suhB*-FLAG₃ (VS066)[12]. *E. coli* MG1655*suhB*(*boxA*(C4T/T6C)), MG1655*suhB*(*boxA*(C4T/T6C))-FLAG₃ and *S.* Typhimurium *hisGΔ*+3::*thyA*, *hisGΔ*+100::*thyA suhB*-FLAG₃ strains were constructed using the 'Flexible Recombineering Using Integration of *thyA*' (FRUIT) method[35].

Plasmids pGB1-pGB36, pGB67–68 were constructed by cloning the *suhB* gene and 200 bp of upstream sequence into the pAMD-BA-*lacZ* plasmid[34], creating transcriptional or translational fusions to *lacZ*. Plasmids pGB192-pGB193 included 200 bp of upstream sequence and 57 nt of *suhB* coding sequence followed by a stop codon. Fusions carrying *boxA* mutations were made by amplifying a *suhB* fragment from GB023 (*boxA*(C4T/T6C)) or by site-directed mutagenesis (*boxA*(C4A)); *suhB* start codon mutations (ATG → CAG) were made using site-directed mutagenesis. Insertions between the *boxA* and S-D sequences were generated by cloning fragments of random non-coding sequence ('GAACTACCCATCTGGTCGCAGATAGTATGAAC'), modified from ref. 36, for insertions of up to 32 bp; 40–100 bp insertions carried a non-coding sequence from the 16S RNA gene in the reverse orientation (region from +1281 to +1380). The 5′ end of the insert remained the same, and inserted sequence was extended towards the S-D sequence (see Supplementary Fig. 6 for details). Plasmid pGB116 was made by cloning the T7 RNAP gene with a S-D sequence into pBAD18[37]. Plasmids pGB83–95 carried the *suhB* gene and 36 nt of the 5′UTR with wt or mutant *boxA*, and a 100 nt insertion between the BoxA and S-D elements, where indicated. *suhB* was under the control of pT7 promoter and was translationally fused to *lacZ* reporter on pAMD-BA-*lacZ* plasmid[34]. Plasmids pGB109–110 were made by cloning CKO_00699(R82A) gene with wt or mutant *boxA* (C4A) and a constitutive

promoter[25]; the toxin gene was transcriptionally fused to *lacZ* reporter on pAMD-BA-*lacZ* plasmid.

**Isolation and identification of *trans*- and *cis*-acting mutants**. The *trans*-acting mutant genetic selection was performed using pJTW067 plasmid carrying a *suhB*-*lacZ* transcriptional fusion in MG1655 Δ*lacZ*. pJTW067 was isolated as an inactive *suhB*-*lacZ* clone from a *cis*-acting mutant screen (see below) and contains a single nucleotide mutation (T607C) that does not affect the expression of *suhB*. It is otherwise identical to pJTW067. Bacterial cultures were grown at 37 °C in LB medium. One hundred microlitres of an overnight culture was washed and plated on M9+0.2% lactose agar. Spontaneous survivors were first tested for increased plasmid copy number using qPCR, comparing the Ct values of plasmid and chromosomal amplicons. Strains with increased copy number were discarded. To eliminate plasmid mutants, plasmids were isolated and transformed into a clean MG1655 Δ*lacZ* background and plated on MacConkey agar indicator plates; mutants forming red colonies (upregulated *suhB*-*lacZ*) were discarded. Chromosomal mutations were identified either by PCR amplification and sequencing of *nusB*, *nusE* and *nusG*, or by whole genome sequencing. Specifically, genomic DNA was purified using a DNeasy blood and tissue kit (Qiagen), DNA libraries were prepared using a Nextera kit (Illumina), sequencing was performed on an Illumina MiSeq instrument, and sequence variants were identified using the CLC genomic workbench (default parameters). The *cis*-acting mutant genetic screen was performed by cloning a mutant *suhB* DNA library, generated by an error-prone DNA polymerase Taq (NEB) with oligonucleotides JW3605 and JW3607, into the pAMD-BA-*lacZ* vector, which was transformed into EPI300 background (*lac*⁻; Epicentre). The mutant library included the entire promoter, 5′ UTR and gene. We selected mutants that were visibly upregulated on MacConkey agar plates and sequenced the insert to identify mutations.

**ChIP-qPCR**. Bacteria were grown at 37 °C in LB medium until OD₆₀₀ = 0.5–0.6. ChIP-qPCR was performed as described previously[34], using monoclonal mouse anti-RpoB (Neoclone #W0002) and M2 monoclonal anti-FLAG (Sigma) antibodies. Occupancy units represent background-subtracted enrichment scores relative to transcriptionally silent regions within the *bglB* or *ynbB* genes in *E. coli*, and the *sbcC* gene in *S.* Typhimurium.

**β-galactosidase assays**. Bacterial cultures were grown at 37 °C in LB medium to an OD₆₀₀ of 0.5–0.6. LB medium was supplemented with 0.2% arabinose when pBAD18 or its derivatives were used. A volume of 100–200 μL of culture was pelleted and resuspended in 800 μL of Z buffer (0.06 M Na₂HPO₄, 0.04 M NaH₂PO₄, 0.01 M KCl, 0.001 M MgSO₄) supplemented with β-mercaptoethanol (50 mM final concentration), sodium dodecyl sulfate (0.001% final concentration), and 20 μL chloroform. Assays were initiated by adding 160 μL of 2-nitrophenyl β-D-galactopyranoside (4 mg/mL) and stopped by adding 400 μL of 1 M Na₂CO₃. The duration of the reaction and OD₄₂₀ readings were recorded and β-galactosidase activity units were calculated as $1000\times (A_{420}/(A_{600})(time_{min}))$.

**Western blotting**. Bacteria were grown at 37 °C in LB to an OD₆₀₀ of 0.5–0.6. Cell pellets were boiled in gel loading dye, separated on gradient polyacrylamide gels (Bio-Rad), and transferred to a polyvinylidene fluoride membrane (Thermo Scientific). The membrane was cut in half at the 50 kDa molecular marker. The upper part of the membrane was probed with control mouse monoclonal anti-RpoC (BioLegend) antibody at 1:4000 dilution, and the lower part was probed with mouse monoclonal M2 anti-FLAG (Sigma) antibody at 1:10,000 dilution. Goat anti-mouse horseradish peroxidase-conjugated antibody was used for secondary probing at 1:20,000 dilution. Both parts of the membrane were always processed in parallel. Blots were developed with Clarity™ Western ECL Substrate (Bio-Rad).

**Sequence alignment of *suhB* upstream regions**. We extracted 100 bp of upstream sequence for *suhB* homologues in 19 species of the family *Enterobacteriaciae*, and aligned the sequences using MUSCLE[38] (Supplementary Fig. 2). To determine the % match to *E. coli* at each position, we added 1 to the number of perfect matches (to account for the *E. coli* sequence), divided by 20 (to account for the 20 species in the alignment), and converted to a percentage.

**Identification of putative *boxA* sequences in γ-proteobacterial genomes**. We searched all sequenced γ-proteobacterial genomes for annotated protein-coding genes with the sequence GYTCTTTAANA within the 50 nt upstream of the annotated gene start. We compared gene functions using COG annotations[39].

**Data availability**. All relevant data supporting the findings of the study are available in this article and its Supplementary Information files. All computer codes are available from the corresponding author upon request.

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

## Acknowledgements

We thank Dave Grainger, Don Court, Todd Gray and Keith Derbyshire for helpful discussions and comments on the manuscript. We thank the Wadsworth Center Bioinformatics and Statistics Core Facility, and the Wadsworth Center Media and Tissue Culture Core Facility for technical assistance. This work was supported by the NIH Director's New Innovator Award Program, 1DP2OD007188 (J.T.W.).

## Author contributions

J.T.W., G.B. and N.S. designed the study. J.T.W. and G.B. wrote the manuscript. G.B., N. S., C.B., R.J., R.F., M.P., A.M.S. and A.S. generated experimental data. P.L. performed bioinformatic analysis. All authors contributed to data analysis and interpretation.

## Additional information

**Competing interests:** The authors declare no competing financial interests.



