## [Peer Review File · Nature Communications]

Reviewers' comments:

Reviewer #1 (Remarks to the Author):

This paper reports evidence that the gene for the transcription factor *suhB* in *E. coli* is autoregulated by a mechanism in which *SuhB*, together with other nus factors, promotes termination of transcription within its gene. This is of considerable interest in that it represents a novel function for nus factors, which are often associated with promoting antitermination rather than with promoting termination. The data for *suhB* gene autoregulation is convincing, but the strong conclusions about the widespread occurrence of this phenomenon are overstatements. Furthermore, the text is too sparse and confusing in places.

1. Abstract. Lines 17-18: This sentence implies that nusB/E (boxA) sites upstream of all nus factor genes serve an autoregulatory function. The data in this paper implicating the boxA sequence upstream of *suhB* in its regulation seems good, and the presence of boxA sites upstream of *suhB* genes in a variety of other genera (Discussion and Table S2) does suggest that *suhB* gene autoregulation may be widespread. However, the idea that all nus factor genes are autoregulated is not supported by data in this paper. Autoregulation of all nus factor genes is mentioned again in the Discussion, with the claim that the evidence for this is strong (p. 12, line 223, and p13, lines 260-261; and again on p. 14, line 270). However, the evidence supporting this seems vague at best. A boxA sequence was noted upstream of a gene adjacent to a nus gene in *Pseudomonas*, with no other evidence to support the relevance of this boxA sequence. These strong conclusions about autoregulation of other nus factor are overinterpretations and should be modified. The idea that all nus factor genes are autoregulated is distinct from the idea that the nus factor complex (including *suhB* itself and the other nus factors) regulates the gene for *suhB*.

2. Discussion, p.13, lines 256-257: The conclusion that the nus factor-boxA regulation mechanism affects a diverse set of genes also seems like an overinterpretation. Aside from *suhB*, where a boxA sequence is frequently found upstream of the gene in many genera, other regulated gene examples seem limited to a couple of genera at best. With the exception of one example shown in Figure 5E, other boxA sequences have not been demonstrated to be functional. What is the probability of finding the boxA consensus sequence in the genome? Are some of these sites random, and not likely functional?

3. p. 9, lines 170-172: These lines describe the reasons for looking at association of *suhB* with a putative boxA in the *hisG* mRNA, but the writing is very confusing. The nature of the previously reported function for the *hisG* boxA is not explained, and the mutation required to abolish translation in order to observe that function is not described. In Figure 5, why was the upstream part of the *hisG* transcript interrupted by *thyA* insertions? Do they somehow mimic the original mutation required to see an effect? It isn't clear that these *hisG* results are interpretable, and they detract from the clear results with the *suhB* gene shown in the same figure. Therefore, perhaps the *hisG* results should not be included in the paper.

4. Discussion, p. 12, lines 232 to end of paragraph. The authors conclude that steric occlusion can't account for the translational repression of the *suhB* gene, since various insertions between the boxA and SD sequences do not prevent autoregulation. However, the insertion of a 100 bp fragment did abolish regulation. What is the interpretation of this result in the context of the loop hypothesis? Would any insertion longer than 100 bp have the same effect, or is this specific to the particular sequence inserted? In the alternative explanation, does NusE bound to NusG necessarily have to be part of the ribosome to inhibit translation of the gene, or could NusE in the context of the ribosome be independent of the nus factor complex?

5. Results, p. 5, last lines on page: The identification of a termination site upstream of ~+400 seems like too specific a conclusion. The nature of the experiment isn't that precise. It looks like the signal strength with the +289 to +416 probe in the nusB and nusE cells is similar to that of

other upstream probes and higher than signal for the next probe downstream (+472 to +611). Does this suggest that the termination event could occur downstream of 416, maybe between 416 and 472 or even a bit beyond 472?

6. P. 6, lines 90-91: The identified mutants are all in boxA, but was a larger region mutagenized? The primers used for the mutagenic PCR are specified, but it would be useful to know what region was tested by this mutational approach.

7. Methods, p. 15: How much *suhB* region sequence is present in the transcriptional and translational fusions (how many nucleotides upstream and downstream of the ORF)? Where are the fusion endpoints with respect to the ORF? Where is the promoter with respect to the *boxA* sequence? Is the putative terminator sequence present in the translational fusion? The information about fusion constructs can presumably be determined from the oligos used to make them, but it would require a lot of effort on the part of the reader (and is too much to expect from a reviewer).

8. Results, p. 6, last paragraph: The specific position of insertions designed to test the steric occlusion is not indicated. Methods, p. 15 says that the 5' end of the insertions "remained the same" and that the inserted sequence was extended towards the S-D element.

9. Results, p.8, line 1: data with a *nusA* mutant is not shown in Figure 3. Is this a mistake?

Reviewer #2 (Remarks to the Author):

The manuscript entitled, "Nus factors have a widespread regulatory function in bacteria", by Baniulyte et al., attempted to establish a role of Nus factors together with the *SuhB* protein in promoting Rho-dependent termination in the *suhB* gene through a translation repression mechanism. They have further tried to show the presence of such type of regulation in two other bacteria.

In my opinion, the work may not be attractive to the general audience of *Nature Communication* rather the manuscript is more suitable for specialized journals like, *Molecular Microbiology*, that too after a thorough revision as per the comments stated or the concerns raised below.

General comments.

1) The observation that Rho-dependent termination occurs in *suhB* is quite interesting and convincing. However, the experimental evidences, on which the claim of Nus factors having roles in promoting this termination event, are not very strong. As stated below in more details, to establish this claim unambiguously, more supporting as well control experiments are required.

2) The authors should have attempted to address mechanistic basis of Nus factors mediated facilitation of the Rho-dependent termination in more details instead of stretching the work into the other bacteria.

3) Sequence analyses to predict the secondary structure around the putative *boxA* sequence as well as identification of a bonafide rut site(s) in the *suhB* is necessary for this manuscript.

4) I would like to see the Nus factors promoting Rho-dependent termination in a purified system by in vitro transcription of the *suhB* gene. Assembling this system in vitro is not technically challenging.

Specific Comments.

1) The title is too general; make it more specific.

2) In Introduction, a reference, *JBC* (2016), 291, 8090-8108 is missing. This paper described inhibitory roles of *NusA* in Rho-dependent termination.

3) Line 61-62; show the upregulation of *suhB* gene in the presence of Rho mutant from the genome-wide analysis as a supplementary data. What are the defects of R66S Rho mutant? Please

show the evidences as supplementary data or mention an appropriate reference where the mutant is described.

4) Line 73; why NusA mutants were not obtained in the screen? NusG mutants should have been included in the analyses. NusG is an integral component of Rho-dependent termination.

5) Lines 77-78; very moderate increase in RNAP occupancy was observed in the presence of Nus mutants as well as boxA mutants. It is better to do in vivo termination assays using reporter system to further reinforce the claims.

6) Lines 81-85; A gradual drop in RNAP occupancy is observed. Where is the rut site? Where is the termination zone? Rho function appears to be weak on this gene. Termination zone may be identified by in vitro transcription assays.

7) Section starting at line 87; is there a proof that suhB interact with boxA sequence? This is not so evident in the previous mBio paper from the same group. It is better to test the occupancy of bona fide boxA binding proteins to prove that this sequence functions as a "boxA."

8) Line 93: Does changes in boxA sequence affect the rut site sequence also? Does this boxA overlap with a rut site? If so, effects of the mutations in the boxA would make rut site weaker that would in turn lead to poor Rho-dependent termination.

9) Line 99; I disagree with the conclusion. The effect is too moderate to make such conclusion. Further experimental support is required.

10) Lines 101-102; Data is not so convincing to make such strong statement.

11) Line 108; simply state SD is not accessible to ribosome. Requirement of NusG for this terminator is not tested. NusB/E association to this putative boxA is also not proved.

12) Line 121; this difference between transcription and translational fusions were not observed for the boxA mutant constructs. Effect of the Rho mutant is same in both these constructs.

13) Line 123; if the inconsistent results are reasoned as a pleiotropic effect, then how can one establish the existence of a specific phenomenon?

14) Figures 3C and D may be omitted.

15) Line 160; the data can still be explained as a steric occlusion of the SD from the ribosome. NusB/E remains bound to boxA even after the RNAP has moved further downstream via suhB (mBio, 2016) –RNAP interaction, where intervening RNA loops out.

16) Line 251: A sequence analysis for the existence of the secondary structures is required.

17) Bioinformatics analyses of the two non-E.coli species appear to be over-stretched. I would prefer to see similar analyses on the E.coli genome to find the distribution of boxA sequences.

18) Line 270; Auto regulation of NusB/NusE genes has not been tested here.

Reviewer #1

“Abstract, Lines 17-18: This sentence implies that nusB/E (boxA) sites upstream of all nus factor genes serve an autoregulatory function. The data in this paper implicating the boxA sequence upstream of suhB in its regulation seems good, and the presence of boxA sites upstream of suhB genes in a variety of other genera (Discussion and Table S2) does suggest that suhB gene autoregulation may be widespread. However, the idea that all nus factor genes are autoregulated is not supported by data in this paper. Autoregulation of all nus factor genes is mentioned again in the Discussion, with the claim that the evidence for this is strong (p. 12, line 223, and p13, lines 260-261; and again on p. 14, line 270). However, the evidence supporting this seems vague at best. A boxA sequence was noted upstream of a gene adjacent to a nus gene in Pseudomonas, with no other evidence to support the relevance of this boxA sequence. These strong conclusions about autoregulation of other nus factor are overinterpretations and should be modified. The idea that all nus factor genes are autoregulated is distinct from the idea that the nus factor complex (including suhB itself and the other nus factors) regulates the gene for suhB.”

There are two lines of evidence that strongly support the idea that Nus factor autoregulation is widespread. First, the *boxA* upstream of *suhB* is widely conserved, and we have shown that it is functional in *Salmonella*. Second, when searching for genes with putative *boxA* sequences <50 bp upstream, only three genes were found in at least three different genera. These genes include *suhB* and *rpsJ* (*nusE*). While this is only circumstantial evidence, it strongly suggests that *nusE* is autoregulated in some species. The finding of a *boxA* sequence upstream of *nusB* in *Pseudomonas* is interesting, but provides only anecdotal evidence for an autoregulatory role of Nus factors. It is also worth noting that we did not find any species in which more than one Nus factor-encoding gene has a putative *boxA* associated with it. In contrast, only 11 of the genera we analyzed lacked a *boxA* upstream of any Nus factor-encoding gene, and at least some of these instances can likely be explained by an altered consensus sequence for BoxA. All this being said, we recognize that we do not have direct experimental evidence for autoregulation of Nus factors beyond *suhB* in *E. coli* and *Salmonella*. Hence, we have softened the language we use to discuss this topic.

“Discussion, p.13, lines 256-257: The conclusion that the nus factor-boxA regulation mechanism affects a diverse set of genes also seems like an overinterpretation. Aside from suhB, where a boxA sequence is frequently found upstream of the gene in many genera, other regulated gene examples seem limited to a couple of genera at best. With the exception of one example shown in Figure 5E, other boxA sequences have not been demonstrated to be functional. What is the probability of finding the boxA consensus sequence in the genome? Are some of these sites random, and not likely functional?”

Our intention was to point out that regulation by Nus factors (other than regulation of rRNA) extends well beyond *E. coli*, and that it extends beyond autoregulation. We do not believe that Nus factors regulate large numbers of genes in any species. We have clarified this point in the manuscript.

“p. 9, lines 170-172: These lines describe the reasons for looking at association of suhB with a putative boxA in the hisG mRNA, but the writing is very confusing. The nature of the previously reported function for the hisG boxA is not explained, and the mutation required to abolish translation in order to observe that function is not described. In Figure 5, why was the upstream part of the hisG transcript interrupted by thyA insertions? Do they somehow mimic the original mutation required to see an effect? It isn't clear that these hisG results are interpretable, and they detract from the clear results with the suhB gene shown in the same figure. Therefore, perhaps the hisG results should not be included in the paper.”

We were concerned that the work on *hisG* in *Salmonella* might appear to detract from the novelty of our study, although there are several reasons to think that the *hisG* BoxA is not functionally relevant (at best, it is only ever used in mutant strains) and most likely not genuine (we detect no Nus factor association with it, and it has

a key mismatch to the consensus sequence). We appreciate that this section is a distraction from the main focus. Nonetheless, we believe it is an important part of the paper. We have opted to include the data as a supplementary figure, with most of the description of the experiment now moved to the figure legend.

*“Discussion, p. 12, lines 232 to end of paragraph. The authors conclude that steric occlusion can’t account for the translational repression of the *suH* gene, since various insertions between the *boxA* and *SD* sequences do not prevent autoregulation. However, the insertion of a 100 bp fragment did abolish regulation. What is the interpretation of this result in the context of the loop hypothesis? Would any insertion longer than 100 bp have the same effect, or is this specific to the particular sequence inserted? In the alternative explanation, does NusE bound to NusG necessarily have to be part of the ribosome to inhibit translation of the gene, or could NusE in the context of the ribosome be independent of the nus factor complex?”*

In light of this comment we revisited this experiment. The construct we previously used was incorrect. We have confirmed that the other constructs are all correct, and remade the reporter fusion with the 100 bp insertion. This fusion behaves similarly to those with shorter insertions. We apologize for the earlier error.

*“Results, p. 5, last lines on page: The identification of a termination site upstream of ~+400 seems like too specific a conclusion. The nature of the experiment isn’t that precise. It looks like the signal strength with the +289 to +416 probe in the *nusB* and *nusE* cells is similar to that of other upstream probes and higher than signal for the next probe downstream (+472 to +611). Does this suggest that the termination event could occur downstream of 416, maybe between 416 and 472 or even a bit beyond 472?”*

We agree with the reviewer that we over-interpreted the ChIP-qPCR data. Moreover, we have performed additional experiments to narrow down the location of the *rut* and termination site. Our new data (Figure 4A) indicate that most Rho-dependent termination occurs within the first 100 bp of the *suH* gene. This is consistent with the ChIP-qPCR data (e.g. compare RNAP association in wild-type and Δ *nusB* cells in the third and fourth regions shown in Figure S1). As discussed in response to a comment from Reviewer #2 (see below), we now believe that the *boxA* and *rut* overlap, and that mutation of the *boxA* disrupts the *rut*. Importantly, our data indicate the presence of additional *rut* sequences that facilitate Rho-dependent termination within *suH* in the *boxA* mutant (Figure 4B). Given that our new data more clearly define the termination signals, and given the difficulty in interpreting the RNAP ChIP-qPCR data, especially in light of the fact that there may be multiple *rut* sequences and multiple termination sites, we have removed the interpretation of these data.

*“P. 6, lines 90-91: The identified mutants are all in *boxA*, but was a larger region mutagenized? The primers used for the mutagenic PCR are specified, but it would be useful to know what region was tested by this mutational approach.”*

We mutagenized the entire promoter, 5' UTR, and gene. This is now stated explicitly in the methods section.

*“Methods, p. 15: How much *suH* region sequence is present in the transcriptional and translational fusions (how many nucleotides upstream and downstream of the ORF)? Where are the fusion endpoints with respect to the ORF? Where is the promoter with respect to the *boxA* sequence? Is the putative terminator sequence present in the translational fusion? The information about fusion constructs can presumably be determined from the oligos used to make them, but it would require a lot of effort on the part of the reader (and is too much to expect from a reviewer).”*

We have added a supplementary figure to show the sequence of the fused regions (Figure S7).

“Results, p. 6, last paragraph: The specific position of insertions designed to test the steric occlusion is not indicated. Methods, p. 15 says that the 5’ end of the insertions “remained the same” and that the inserted sequence was extended towards the S-D element.”

This information is now provided in Figure S7.

“Results, p.8, line 1: data with a nusA mutant is not shown in Figure 3. Is this a mistake?”

This is a mistake, and has been removed.

Reviewer #2

*“The observation that Rho-dependent termination occurs in *suhB* is quite interesting and convincing. However, the experimental evidences, on which the claim of Nus factors having roles in promoting this termination event, are not very strong. As stated below in more details, to establish this claim unambiguously, more supporting as well control experiments are required.”*

We respectfully disagree. We believe that we have provided irrefutable evidence of Nus factor regulation of Rho-dependent termination for *suhB*. Multiple, independent lines of evidence support this conclusion: (i) we have shown BoxA-dependent physical association of Nus factors with transcription complexes, (ii) random mutagenesis of the entire *suhB* gene and upstream region yielded termination-defective mutants that all have mutations in a conserved sequence that perfectly matches the *boxA* consensus; (iii) isolation of spontaneous *trans*-acting mutants that disrupt Rho termination of *suhB* yielded mutants only of Nus factor-encoding genes; (iv) multiple, independent, targeted assays confirm the effects of the putative BoxA and Nus factors on Rho-dependent termination within *suhB*.

“The authors should have attempted to address mechanistic basis of Nus factors mediated facilitation of the Rho-dependent termination in more details instead of stretching the work into the other bacteria.”

We are very interested in the mechanism of Nus factor function at *suhB*, and at other loci. However, we feel that additional work on this topic is beyond the scope of the current study. Indeed, it is likely that we are still many years from a complete mechanistic understanding of Nus factor function.

*“Sequence analyses to predict the secondary structure around the putative *boxA* sequence as well as identification of a bonafide *rut* site(s) in the *suhB* is necessary for this manuscript.”*

We have now localized the termination signals within the region up to 57 bp into *suhB* (Figure 4C). Moreover, our new data strongly suggest that the *rut* and *boxA* sequences overlap, such that mutation of the *boxA* disrupts the *rut*. Specifically, we found that a short transcriptional fusion of *suhB* with upstream sequence and only 57 bp of the gene, is Rho-terminated in a BoxA-dependent manner (Figure 4A). An equivalent construct in which the *suhB* start codon was mutated is also Rho-terminated. This is expected, since preventing translation of *suhB* by mutating the start codon should be at least as effective as BoxA-mediated repression. Unexpectedly, simultaneous mutation of both the *boxA* and the *suhB* start codon substantially reduced Rho-dependent termination (Figure 4A), strongly suggesting that the *boxA* mutation disrupts the *rut*. The *boxA* sequence is the most pyrimidine-rich sequence within the *suhB* 5’ UTR – a 21 bp sequence overlapping the *boxA* has 17 pyrimidines and only one G – and *rut* sequences are known to be pyrimidine-rich and G-poor. Thus, our experimental data suggesting that the *boxA* and *rut* overlap are supported by sequence analysis. We have also

shown that there is at least one additional *rut* within the *suhB* gene. Specifically, mutation of both the *suhB* start codon and the *boxA* does not disrupt Rho-dependent termination for a longer transcriptional fusion that includes the entire *suhB* gene (Figure 4B). This is important because we have other data for constructs containing a mutated *boxA*, but all these constructs include the entire *suhB* gene.

“I would like to see the Nus factors promoting Rho-dependent termination in a purified system by in vitro transcription of the suhB gene. Assembling this system in vitro is not technically challenging.”

Assembling an *in vitro* system to study regulation of *suhB* transcription would require purification of at least six proteins (NusB, NusE, NusA, NusG, SuhB and Rho), and perhaps also ribosomal proteins S4, L3, L4 and L13 that have been implicated in the antitermination process. Maximal antitermination complex activity has never been described in the literature using a fully purified system. This might be due to the lack of SuhB, but equally could indicate that there are additional as-yet-unidentified Nus factors. As much as we would like to study Nus factor function with an *in vitro* system, it is simply not practical to do so.

“The title is too general; make it more specific.”

We have changed the title to “Identification of regulatory targets for the bacterial Nus factor complex”.

“In Introduction, a reference, JBC (2016), 291, 8090-8108 is missing. This paper described inhibitory roles of NusA in Rho-dependent termination.”

We have added a citation for this paper.

“Line 61-62; show the upregulation of suhB gene in the presence of Rho mutant from the genome-wide analysis as a supplementary data. What are the defects of R66S Rho mutant? Please show the evidences as supplementary data or mention an appropriate reference where the mutant is described.”

We have added a reference to a paper that described characterization of this mutant.

“Line 73; why NusA mutants were not obtained in the screen? NusG mutants should have been included in the analyses. NusG is an integral component of Rho-dependent termination.”

We do not know why NusA mutants were not obtained in the screen. The same applies to Rho mutants. We note that NusG mutants were isolated far less frequently than NusB mutants or NusE mutants. We excluded NusG mutants from the later analyses because it is unclear whether these mutants are simply defective in Rho-dependent transcription termination, or in Nus complex function; we have unpublished data to indicate that at least two of the three NusG mutants are at least partially defective in Rho-dependent termination. It is important to note that there are no described NusG mutants that are specifically defective in Nus complex function. NusG was first identified as a Nus factor by virtue of mutants that suppress the defects of other Nus factor mutants.

“Lines 77-78; very moderate increase in RNAP occupancy was observed in the presence of Nus mutants as well as boxA mutants. It is better to do in vivo termination assays using reporter system to further reinforce the claims.”

The increase in RNAP occupancy is moderate but reproducible. The overall autoregulatory phenomenon we observe for *suhB* is not huge, but our data are nonetheless convincing. Importantly, the *lacZ* fusion assays provide independent evidence for Nus factor-mediated Rho termination, and get around the potential problem of pervasive transcription from the chromosome in the Rho mutant.

“Lines 81-85; A gradual drop in RNAP occupancy is observed. Where is the rut site? Where is the termination zone? Rho function appears to be weak on this gene. Termination zone may be identified by in vitro transcription assays.”

As described above, we have new data suggesting that the *boxA* and *rut* sites overlap, and indicating Rho-dependent termination upstream of position 57 of the *suhB* gene.

*“Section starting at line 87; is there a proof that *suhB* interact with *boxA* sequence? This is not so evident in the previous mBio paper from the same group. It is better to test the occupancy of bona fide *boxA* binding proteins to prove that this sequence functions as a “*boxA*.””*

We do not believe that SuhB binds directly to the BoxA. Nonetheless, our earlier study in *mBio* clearly showed that SuhB association with transcription complexes is dependent upon NusB association with a functional BoxA.

*“Line 93: Does changes in *boxA* sequence affect the *rut* site sequence also? Does this *boxA* overlap with a *rut* site? If so, effects of the mutations in the *boxA* would make *rut* site weaker that would in turn lead to poor Rho-dependent termination.”*

As discussed above, we have included new data that strongly suggest the *rut* and *boxA* sequences overlap (Figure 4A), as the reviewer proposes. However, the data in Figure 4B demonstrate that additional *rut* elements within *suhB* can cause efficient Rho-dependent termination in all of the other constructs used that have a mutated *boxA*.

“Line 99; I disagree with the conclusion. The effect is too moderate to make such conclusion. Further experimental support is required.”

We respectfully disagree. The increase in RNAP occupancy is moderate but reproducible, and supported by our *lacZ* fusion experiments.

“Lines 101-102; Data is not so convincing to make such strong statement.”

As described above, we believe our data provide convincing evidence of Nus factors promoting Rho-dependent transcription termination within *suhB*.

*“Line 108; simply state SD is not accessible to ribosome. Requirement of NusG for this terminator is not tested. NusB/E association to this putative *boxA* is also not proved.”*

We are unsure what the reviewer means. We hypothesize that NusB/E association with the BoxA directly occludes the Shine-Dalgarno sequence. Later in the manuscript we modify this hypothesis based on data showing that simple occlusion does not explain translational repression of *suhB* by Nus factors.

“Line 121; this difference between transcription and translational fusions were not observed for the boxA mutant constructs. Effect of the Rho mutant is same in both these constructs.”

We would not expect to see Rho termination with the *boxA* mutant constructs since these no longer recruit Nus factors.

“Line 123; if the inconsistent results are reasoned as a pleiotropic effect, then how can one establish the existence of a specific phenomenon?”

The *boxA* mutant would not be subject to pleiotropic effects, and this confirms the result with Nus factor mutants.

“Figures 3C and D may be omitted.”

We prefer to include these figure panels because they demonstrate relevance in the native context.

*“Line 160; the data can still be explained as a steric occlusion of the SD from the ribosome. NusB/E remains bound to boxA even after the RNAP has moved further downstream via *suhB* (mBio, 2016) –RNAP interaction, where intervening RNA loops out.”*

We agree. Specifically, the RNA loop would be too small for the ribosome to access the Shine-Dalgarno sequence. We have clarified this in the text.

“Line 251: A sequence analysis for the existence of the secondary structures is required.”

We have not included a secondary structure prediction because (i) these predictions are often inaccurate, and (ii) it is unclear what length of RNA to use for such a prediction.

*“Bioinformatics analyses of the two non-*E. coli* species appear to be over-stretched. I would prefer to see similar analyses on the *E. coli* genome to find the distribution of boxA sequences.”*

A variety of *E. coli* strains, including MG1655, were analyzed (Table S3).

“Line 270; Auto regulation of NusB/NusE genes has not been tested here.”

As described in the response to Reviewer #1, we believe that the evidence for NusE autoregulation is strong. Nonetheless, we have softened the language in this section.

REVIEWERS' COMMENTS:

Reviewer #1 (Remarks to the Author):

Comments about the responses to our review (reviewer 1) of NCOMMS-16-29895A; Wade and colleagues:

Author response #1:

The authors indicate that in their revised manuscript they have softened their interpretation of the statement that Nus factor autoregulation is widespread. However, this conclusion is still being stated too strongly (or too imprecisely). See revised abstract, lines 18-22, where it is now said that "extensive conservation of NusB/E binding sites upstream of nus factor genes suggests that Nus factor autoregulation occurs". The problem is that this statement still implies that there is extensive conservation of NusB/E binding sites upstream of each of the other Nus factor genes in addition to *suhB*. However, not only do they not have any experimental evidence for this, but their analysis of conservation of sequences doesn't even support that this is a widespread phenomenon. The statement still seems very misleading. See also lines 55-56, which (over)state that autoregulation of *nusE* or *nusB* as well as *suhB* is a common phenomenon.

I think that the concept the authors are trying to get across is that the activity of the Nus factor complex is likely to be affected by regulation of expression of *SuhB* by the Nus factor complex in a wide variety of bacteria. This is different from saying that expression of each of the Nus factor genes is directly regulated by the Nus factor complex in most bacteria. This distinction should be made.

Author response #2:

The authors clarified the point that Nus factors do not regulate a large number of genes in any given species (Discussion, lines 305-307). However, the Abstract, lines 19-20, still implies that they do. "Putative NusB/E binding sites are also found upstream of many other genes in diverse species...." This statement should also be modified.

Author response #3:

The authors agreed that the data about the BoxA sequence upstream of *hisG* gene detracted from the paper, and they removed the Figure from the main text. Their response says..."there are several reasons to think that the *hisG* BoxA is not functionally relevant (at best, it is only used in mutant strains), and most likely not genuine (we detect no Nus factor association with it, and it has a key mismatch in the consensus sequence....).

However, to address previously published reports that this BoxA was functional, a very abbreviated statement about the experiment is now included in the main text, and the *hisG* experimental data is shown as a Supplemental Figure (S6). This seems like a reasonable solution.

Author response #4: An error was found in a strain and issues were resolved.

Author response #5: Additional experiments were carried out to address the approximate location of the termination site and are reported in the revised Results section text, lines 139-175, and in Figure 4. (However, the authors should probably cite Figure S7 in the text for the sequence of the *suhB* region.) The previously included ChIP-qPCR data addressing the location of the termination site were not interpretable and have now been removed. The conclusion that there may be additional termination sites within the gene seems reasonable and consistent with the new findings.

Author response #6: Clarified as requested.

Author response #7: Supplementary Figure S7 was added to provide the information requested,

and is extremely useful and important.

Author response #8: Mistake in original text was corrected.

Reviewer #2 (Remarks to the Author):

The authors have included many of the suggestions and the responded to the comments made by this referee. The manuscript has improved considerably. However, I still have the following concerns.

1) Figure 1: The increase in the RNAP occupancy at the end of the *suhB* in the presence of the Nus mutants compared to the Rho mutant is very moderate. RNAP occupancy at the beginning of the gene is even less pronounced both in the presence of the Nus and the Rho mutants (figure S1), indicating that the "terminator" present before the gene is very weak.

2) Figure 3: Increase in the beta galactosidase activity at the beginning of the gene in the presence of the Rho mutant does not correlate with the massive increase in the RNAP occupancy at the end of the gene, indicating the presence of an another terminator inside the *suhB* that is also proposed by the authors. I feel that the terminator present inside the gene plays the important role in terminating the *suhB* and not the weak one present at the 5'-UTR. The terminator present inside the gene does not have an overlapping boxA sequence and is not Nus factor-dependent.

3) In my opinion, the data indicating the presence of direct translation repression by the Nus factors is very convincing, and the authors should emphasize that the translation repression is the major mechanism guiding the Nus factor-dependent regulation of this gene.

Response to Reviewer #1

“Author response #1:

*The authors indicate that in their revised manuscript they have softened their interpretation of the statement that Nus factor autoregulation is widespread. However, this conclusion is still being stated too strongly (or too imprecisely). See revised abstract, lines 18-22, where it is now said that “extensive conservation of NusB/E binding sites upstream of nus factor genes suggests that Nus factor autoregulation occurs”. The problem is that this statement still implies that there is extensive conservation of NusB/E binding sites upstream of each of the other Nus factor genes in addition to *suhB*. However, not only do they not have any experimental evidence for this, but their analysis of conservation of sequences doesn’t even support that this is a widespread phenomenon. The statement still seems very misleading. See also lines 55-56, which (over)state that autoregulation of *nusE* or *nusB* as well as *suhB* is a common phenomenon.*

*I think that the concept the authors are trying to get across is that the activity of the Nus factor complex is likely to be affected by regulation of expression of *SuhB* by the Nus factor complex in a wide variety of bacteria. This is different from saying that expression of each of the Nus factor genes is directly regulated by the Nus factor complex in most bacteria. This distinction should be made.”*

The point we are trying to make is that most gammaproteobacterial species appear to have Nus factor autoregulation. This is usually through regulation of *suhB*, but our sequence analysis strongly suggests that *rpsJ* (encodes NusE) is autoregulated in multiple genera, and we provide circumstantial evidence that *nusB* may also be autoregulated in some species. Moreover, in about half of the genera where we did not find a putative *boxA* upstream of *suhB* (excluding genera where we failed to find a *boxA* upstream of rRNA loci), there is a putative *boxA* upstream of *rpsJ* or *nusB*. This suggests that one Nus factor-encoding gene is autoregulated in the vast majority of species. Interestingly, we found no species that appear to have more than one autoregulated Nus factor-encoding gene. Given the word limit of the abstract we have chosen to simply remove the word “extensive”. We have clarified the text in the introduction to indicate that *suhB* is the predominant regulatory target in gammaproteobacterial.

“Author response #2:

The authors clarified the point that Nus factors do not regulate a large number of genes in any given species (Discussion, lines 305-307). However, the Abstract, lines 19-20, still implies that they do. “Putative NusB/E binding sites are also found upstream of many other genes in diverse species....” This statement should also be modified.”

We have altered the text to say “many putative NusB/E binding sites are found upstream of other genes in diverse species”. We think this is a fair reflection of the bioinformatic analysis that identified putative *boxA* sequences upstream of >150 different genes in other species.

Response to Reviewer #2

“The authors have included many of the suggestions and the responded to the comments made by this referee. The manuscript has improved considerably. However, I still have the following concerns.

*1) Figure 1: The increase in the RNAP occupancy at the end of the *suhB* in the presence of the Nus mutants compared to the Rho mutant is very moderate. RNAP occupancy at the beginning of the gene is even less pronounced both in the presence of the Nus and the Rho mutants (figure S1), indicating that the “terminator” present before the gene is very weak.”*

Interpretation of the ChIP-qPCR data is confounded by the fact that signal “bleeds through” from adjacent regions. ChIP-enriched DNA fragments extend into adjacent regions, and many are likely to include sufficient sequence to be amplified by upstream/downstream PCR primer pairs. We have chosen to limit our interpretation of the ChIP-seq data, especially in light of the data generated using alternative approaches, which together strongly support the existence of a Rho termination site early in *suhB*.

“2) Figure 3: Increase in the beta galactosidase activity at the beginning of the gene in the presence of the Rho mutant does not correlate with the massive increase in the RNAP occupancy at the end of the gene, indicating the presence of an

*another terminator inside the *suhB* that is also proposed by the authors. I feel that the terminator present inside the gene plays the important role in terminating the *suhB* and not the weak one present at the 5'-UTR. The terminator present inside the gene does not have an overlapping *boxA* sequence and is not Nus factor-dependent."*

We agree with the reviewer that our data support the existence of an additional, Nus factor-independent Rho termination site. Nonetheless, we believe our data strongly support a role for Nus factors in controlling transcription of *suhB* through Rho termination, albeit as a "back-up" to translational regulation.

"3) In my opinion, the data indicating the presence of direct translation repression by the Nus factors is very convincing, and the authors should emphasize that the translation repression is the major mechanism guiding the Nus factor-dependent regulation of this gene."

We agree with the reviewer that translational repression is the primary mechanism of regulation by Nus factors. Nonetheless, our data strongly support an additional level of regulation, i.e. Rho-dependent transcription termination, which would be expected in the presence of translational repression. We have modified the section title in the Results where we introduce Nus factor regulation of *suhB* to focus solely on translational repression. We have also modified the abstract to say that "repression occurs primarily by translation inhibition".